# The Antimicrobial and Toxicity Influence of Six Carrier Oils on Essential Oil Compounds

**DOI:** 10.3390/molecules28010030

**Published:** 2022-12-21

**Authors:** Salehah Moola, Ané Orchard, Sandy van Vuuren

**Affiliations:** Department of Pharmacy and Pharmacology, Therapeutic Sciences, University of the Witwatersrand, Johannesburg 2193, South Africa

**Keywords:** antimicrobial, toxicity, carrier oils, essential oil compound, minimum inhibitory concentration, selectivity index, ESKAPE, percentage mortality, thymoquinone, *Prunus armeniaca*, *Calendula officinalis*

## Abstract

Essential oil compounds have been identified as alternative antimicrobials; however, their use is limited due to their toxicity on human lymphocytes, skin, and reproduction. Carrier oils can reduce the toxicity of essential oils, which raises the question as to whether such activity would extend to the essential oil compounds. Thus, this study aimed to investigate the antimicrobial and toxicity activity of essential oil compounds in combination with carrier oils. The antimicrobial properties of the essential oil compounds, alone and in combination with carrier oils, were determined using the broth microdilution assay. The toxicity was determined using the brine shrimp lethality assay. Antimicrobial synergy (ΣFIC ≤ 0.50) occurred in 3% of the samples when tested against the ESKAPE pathogens. The compound thymoquinone in combination with the carrier oil *Prunus armeniaca* demonstrated broad-spectrum synergistic activity and a selectivity index above four, highlighting this combination as the most favorable. The carrier oils reduced the toxicity of several compounds, with *Calendula officinalis* and *P. armeniaca* carrier oils being responsible for the majority of the reduced toxicity observed. This study provides insight into the interactions that may occur when adding a carrier oil to essential oil compounds.

## 1. Introduction

Antimicrobial resistance is responsible for a large number of morbidities and mortalities worldwide [1,2]. The World Health Organization published a list of priority micro-organisms known as the ESKAPE pathogens (*Enterococcus faecium*, *Staphylococcus aureus*, *Klebsiella pneumoniae*, *Acinetobacter baumannii*, *Pseudomonas aeruginosa*, and *Escherichia coli*) which requires urgent attention with respect to antimicrobial resistance [3]. Essential oils have been identified as alternative antimicrobial options due to their observed antimicrobial properties against a wide range of pathogens, especially against Gram-positive micro-organisms [4,5,6,7,8,9,10].

Essential oils display various biological activities that include antimicrobial [11,12,13,14], insecticidal [11,15,16], food preservative [12,17,18,19], anti-oxidative [14,19], anti-inflammatory [14,20] and anticancer [21,22] activities. Essential oils comprise anywhere from 10 to more than 300 compounds belonging to many different chemical classes such as alcohols, oxides or ethers, aldehydes, esters, ketones, amides, amines, heterocycles, phenols, and terpenes [23]. These compounds possess antimicrobial activity, with different compounds such as carvacrol, cinnamaldehyde, eugenol, geraniol, and thymol exhibiting varying degrees of activity against pathogens such as *S. aureus* and *E. coli* [4,6,23,24,25]. Despite the extensive use of these compounds and the plethora of studies reporting their antimicrobial potential, their use in humans is limited by their toxicity which has been shown against human lymphocytes, hepatocytes, skin, reproduction, and mucous membranes [26,27,28,29].

Carrier oils, also known as fixed oils, are made of a number of lipids such as waxes or fatty acids (Omega 3 and 6) as well as vitamins (E and A) and minerals [30]. These are produced by methods of centrifugation, maceration, cold press, or extraction from the fatty component of a plant [30]. Carrier oils have been shown to reduce the toxicity of essential oils [31]. The constituents responsible for the reduction in toxicity is not known; however, the carrier oils often contain vitamin E, and studies reporting the decrease in toxicity of toxic medicines by vitamin E are available [31,32,33].

A previous study [31], investigated the combinations of carrier oils with essential oils; however, it was not known whether the previous observations would extend to the single compounds that are found within essential oils. The hypothesis of whether the synergy exerted by the carrier oils on essential oils would extend to the essential oil compounds thus arose. Therefore, this study explored both the antimicrobial and toxicity interactions between a selection of essential compounds and carrier oils to determine which combinations would provide the optimum antimicrobial combination with the least toxicity.

It has been recommended that in studies which investigate bioactive preparations from natural sources, the selectivity index should be examined. This is important in order to determine a safe therapeutic dose which is still active [34]. Taking this into consideration, the selectivity index was determined in optimum combinations, providing insight into favorable combinations.

## 2. Results and Discussion

### 2.1. Antimicrobial Analysis

Poor antimicrobial activity was mostly observed with the six carrier oils (*Aloe vera, Calendula officinalis, Hypericum perforatum, Persea americana, Prunus armeniaca*, and *Simmondsia chinensis*) selected for the study (Table 1). Some noteworthy activity was displayed against *C. albicans* by *A. vera*, *C. officinalis* and *S. chinensis*. Minimal antimicrobial activity was expected from the carrier oils due to their consisting of vitamins, minerals, and free fatty acids, which are not known for antimicrobial activity [30]. The lack of antimicrobial activity of the carrier oils has also been previously noted [31]. It was, however, important to document the minimum inhibitory concentration (MIC) values as it forms an important starting point for determining the combined fractional inhibitory concentration index (ƩFIC) values.

Several compounds tested displayed noteworthy antimicrobial activity (≤1.00 mg/mL) against all pathogenic reference strains tested (Table 2). These compounds included: carvacrol, cinnamaldehyde, isoeugenol, and thymol. This highlights the antimicrobial importance of these compounds, since they were able to maintain broad-spectrum, noteworthy antimicrobial activity across all the strains tested. Previous studies have also reported on the antimicrobial activities of carvacrol, cinnamaldehyde, isoeugenol, and thymol against several of the pathogens [35,36,37,38,39,40,41,42,43,44,45,46]. Phenolic compounds (carvacrol and thymol) could be potential agents to fight against antimicrobial resistance as data has shown that phenolics inhibit resistant strains [47,48,49].

#### Combinations

The MIC values of the 21 compounds in combination with six carrier oils against seven pathogens (882 combinations in total) were determined, and the results are shown in Table 3, Table 4, Table 5, Table 6, Table 7, Table 8 and Table 9. In summary, most of the combinations resulted in indifference (56%), followed by 37% additive interactions. There was a total of 3% synergistic (23 combinations) and 4% antagonistic interactions noted.

The combined use of *A. vera, C. officinalis*, and *H. perforatum* with α-terpinene resulted in the most synergy against *E. faecium* (ƩFIC value of 0.41) (Table 3). The compound α-terpinene was present in 75% of the synergistic interactions against *E. faecium*. The combined use of thymoquinone with *H. perforatum* resulted in the most antagonistic interaction against *E. faecium* (ƩFIC value of 12.01), and the compound thymoquinone was present in 83% of the antagonistic interactions against *E. faecium*.

Against *S. aureus*, the combination of thymoquinone with *H. perforatum* resulted in the most synergistic interaction (ƩFIC value of 0.13), and the compound thymoquinone was present in most of the synergistic interactions (46%), displaying synergy when combined with all six carrier oils tested (Table 4). The combination of santalol with *A. vera* displayed the highest antagonistic interaction against *S. aureus* (ƩFIC value of 8.75).

None of the compound: carrier oil combinations displayed synergistic interactions against *K. pneumoniae*, and one combination (santalol with *P. americana*) displayed antagonism (ƩFIC value of 6.00) (Table 5).

None of the compound: carrier oil combinations resulted in synergy against *A. baumannii* (Table 6). Antagonism was apparent for combinations of α-terpinene with *P. americana*, and (+)-α-pinene with *P. armeniaca* (ƩFIC values of 6.67). The carrier oils *P. americana* and *P. armeniaca* were present most often in antagonistic interactions (40%).

None of the compound: carrier oil interactions displayed synergistic interactions against *P. aeruginosa* (Table 7). The combination of *p*-cymene with *A. vera* was the only combination which resulted in antagonism (ƩFIC value of 7.24).

The combination demonstrating the most synergy against *E. coli* was thymoquinone and *A. vera* (ƩFIC value of 0.09) (Table 8). Thymoquinone was present in all of the synergistic interactions observed. The combinations which showed the most antagonism was *p*-cymene combined with *S. chinensis* and linalool combined with *S. chinensis* (ƩFIC values of 4.95). The compound santalol (29%) and carrier oil *S. chinensis* (57%) were present most often in antagonistic interactions.

The combination of *p*-cymene with *H. perforatum* and β-caryophyllene with *P. armeniaca* resulted in synergistic interactions against *C. albicans* (ƩFIC values of 0.50) (Table 9). None of the combinations resulted in antagonism.

In summary (Figure 1), the compound thymoquinone and the carrier oil *P. armeniaca* were present in the majority of the synergistic combinations. The carrier oil *H. perforatum* and the compound santalol were present most frequently in the antagonistic combinations.

Of the four compounds (carvacrol, cinnamaldehyde, isoeugenol, and thymol) which showed broad-spectrum, noteworthy antimicrobial activity against all reference strains tested, only thymol produced some synergistic antimicrobial activity when combined with carrier oils. This suggests that noteworthy antimicrobial activity of a compound by itself does not necessarily correlate to synergy when combined with carrier oils. In fact, the compounds cinnamaldehyde, citral, santalol, and thymoquinone, which showed noteworthy MIC values when tested alone, were present in several antagonistic combinations when combined with carrier oils. Thymoquinone and santalol in particular were present repeatedly in antagonistic combinations against more than one reference strain. The influence of the carrier oil on the antimicrobial activity of a compound differed according to the reference strain tested.

When observing the interactive profiles of the compound: carrier oil combinations against the various reference strains tested, it was noted that the combinations tested against the Gram-positive bacteria displayed the highest synergy (6%) as well as the highest antagonism (6%). The combinations tested against the Gram-negative bacteria displayed the least synergy (1%), and combinations tested against the yeast reference strain displayed the second highest synergy (2%) and the least antagonism (0%). Thymoquinone was the compound most commonly observed in the synergistic antimicrobial interactions.

In another study [31] where essential oils were combined with carrier oils against skin pathogens, most of the synergistic interactions also occurred against the Gram-positive bacteria. The enhanced susceptibility of the Gram-positive bacteria may be due to the susceptibility of the outer membrane where the structure is less complex than that of the Gram-negative micro-organisms. The structure consists of a membrane weaker than that of the Gram-negative bacteria and consists only of a thick peptidoglycan wall which is not adequate to prevent the entry of antimicrobial compounds [50,51].

The antimicrobial enhancing properties of the carrier oils present in the synergistic combinations could be attributed to their free fatty acids [31]. Free fatty acids, such as oleic and linoleic acid, have been reported to show antimicrobial activity against the Gram-positive micro-organism *S. aureus* when tested at high concentrations [52]. The antimicrobial activity of free fatty acids could be attributed to their ability to cause cell lysis, disruption to nutrient uptake, inhibition of enzyme activity, and formation of auto-oxidation products, as well as their ability to alter pH levels, thus causing disturbance to the bacterial membrane [53,54].

In a previous study [31], combinations of essential oils with the carrier oils *A. vera*, *H. perforatum*, *P. americana*, *P. armeniaca* and *S. chinensis* resulted in synergy against *C. albicans*. In this study, combinations of two compounds with *H. perforatum* and *P. armeniaca* resulted in synergy. The two compounds which showed synergy against *C. albicans* were present as major compounds in the essential oils *Kunzea ericoides* A.Rich. Joy Tomps. (kanuka) and *Lavandula angustifolia* Mill. (lavender) which also resulted in synergy with the carrier oils [31]. This demonstrates that there were instances where the synergy observed for a single compound: carrier oil combination correlated with the synergistic interaction observed by the neat essential oil: carrier oil combination. As an example, *K. ericoides* essential oil, containing *p*-cymene (11.9%) as a major compound, resulted in synergy when combined with *A. vera*, and *p*-cymene combined with *A. vera* resulted in synergy in this study. This was also observed with *L. angustifolia* essential oil, containing linalyl acetate (35.6%), which resulted in synergy when combined with *C. officinalis;* and in this study, the combination of linalyl acetate with *C. officinalis* also resulted in synergy.

Very few synergistic interactions were observed against the Gram-negative bacteria previously [31], and in this study. The combinations which did result in synergy against Gram-negative bacteria were thymoquinone combined with *A. vera, P. americana, P*. *armeniaca,* and *S. chinensis.* Antagonism was seen most frequently against the Gram-negative bacteria by the compound santalol and the carrier oils *P. americana* and *S. chinensis*.

### 2.2. Toxicity Analysis

All six carrier oils tested were non-toxic at both 24 and 48 h (Table 10). The least toxic of the carrier oils was *P. americana*. These results are congruent with a previous carrier oil study [31].

At 24 h, 24% of the compounds showed non-toxic results and 19% of the compounds showed non-toxic results at 48 h. At both 24 and 48 h, the compounds β-caryophyllene, p-cymene, linalyl acetate, and γ-terpinene were non-toxic, and R (+)-limonene was non-toxic only at 24 h. The compounds p-cymene (at 24 h), linalyl acetate, and γ-limonene showed non-toxicity in previous studies [55,56,57]. The other compounds (76% and 81%) showed toxicity to the brine shrimp either at 24 h or, at both 24 and 48 h, showing the highly toxic nature of the compounds by themselves, even when diluted to a concentration of 1.00 mg/mL.

The compounds carvacrol, citral, eugenol, cinnamaldehyde, geraniol, linalool, menthol, nerol, α-pinene, santalol, γ-terpinene, terpinene-4-ol, thymol, and thymoquinone have previously demonstrated various biological toxicities [58,59,60,61,62,63,64,65,66,67,68,69]. This study, together with the literature, shows that the essential oil compounds tested are predominantly toxic, and unless a means of decreasing their toxicity is found, their application for humans is limited.

#### Combinations

After combining the 21 compounds with all six carrier oils (Table 11, Table 12, Table 13, Table 14, Table 15 and Table 16), it was found that in several instances the toxicity of the compounds was reduced. At 24 h, the combinations containing *C. officinalis*, *H. perforatum*, and *P. armeniaca* resulted in the most reduction in compound toxicity, and at 48 h, *H. perforatum* resulted in the most reduction in compound toxicity. The carrier oil *H. perforatum* would therefore be a suitable option to be combined with compounds tested for the purpose of reducing toxicity.

The combination of *p*-cymene with *A. vera* resulted in the most favorable synergistic interaction at 24 h (ƩFIC value of 0.28) (Table 11), and α-terpinene when combined with *A. vera* resulted in the only synergistic interaction observed at 48 h. The highest antagonistic ƩFIC values at 24 and 48 h resulted from the combination of γ-terpinene with *A. vera* (ƩFIC values of 49.32 and 18.01 respectively).

At 24 h, when R (+)-limonene and γ-terpinene were combined with *C. officinalis*, a complete reduction in toxicity was observed (Table 12). γ-Terpinene combined with *C. officinalis* resulted in the only synergistic interaction observed at 48 h (ƩFIC value of 0.25). Several compound: carrier oil combinations resulted in the most antagonistic interactions observed at both 24 and 48 h (ƩFIC value of 44.36).

The combined use of santalol and *H. perforatum* was the only synergistic interaction at 24 h, and at 48 h γ-terpinene and *H. perforatum* resulted in the most synergistic interaction (ƩFIC value of 0.42) (Table 13). Several compound: carrier oil combinations resulted in antagonistic ƩFIC values at 24 and 48 h.

The toxicity of *P. americana* alone at 24 h was 0.00% and so ƩFIC values at 24 h could not be calculated; however, it could be noted that the toxicity of the compounds isoeugenol, linalool, santalol, α-terpinene, and (+)-terpinen-4-ol reduced from toxic levels to non-toxic levels when combined with *P. americana* at 24 h. None of the compound: carrier oil combinations displayed synergy at 48 h (Table 14). The combination of γ-terpinene and *H. perforatum* was the most antagonistic (ƩFIC value of 54.52).

At 24 h, the combination of santalol and *P. armeniaca* resulted in a complete decrease in toxicity, followed by R (+)-limonene and *P. armeniaca* which resulted in the second most synergistic interaction with an ƩFIC value of 0.20. At 48 h, santalol or R (+)-limonene combined with *P. armeniaca* resulted in the most synergistic interactions (ƩFIC values of 0.19). At 24 and 48 h, several combinations were antagonistic (Table 15).

All of the compounds with *S. chinensis* resulted in antagonism at 24 h (Table 16). Less antagonism was observed at 48 h, where the most antagonistic ƩFIC value resulted from linalyl acetate combined with *S. chinensis* (ƩFIC value of 4.89).

Table 17 provides a summary of the toxicity percentage of the interactions of each carrier oil in combination with the essential oil compounds. Synergy indicates that the carrier oil was able to quench the toxicity of the essential oil compounds, rendering it non-toxic. The carrier oil *P. armeniaca* resulted in the most synergy in its respective combinations with the compounds at 48 h. A constituent of *P. armeniaca*, vitamin E [31], may be the contributing factor to the carrier oil’s favorable toxicity quenching abilities as it was previously reported that vitamin E was able to reduce the toxic effect of the medicine digoxin in rabbits [32] and acute mercury toxicity in rats [70].

At 24 h, the carrier oil *S. chinensis* resulted in the most antagonism within its combinations, and *C. officinalis* (responsible for majority of the synergistic interactions) showed the least antagonism. Therefore, at 24 h, *C. officinalis* would be the most favorable carrier oil choice to be combined with the compounds used in this study to reduce their toxicity. At 48 h, the carrier oil *P. americana* was responsible for the majority of the antagonistic interactions, and *S. chinensis* showed the least.

The compounds that most commonly quenched toxicity and therefore resulted in synergistic interactions when combined with the carrier oils at both 24 and 48 h were α-terpinene; linalyl acetate; γ-terpinene; R (+)-limonene; and santalol. The compound R (+)-limonene quenched toxicity the most.

To the best of our knowledge, to date there have been no previous studies conducted on the toxicity of the combined use of essential oil compounds with carrier oils; however, there has been a study on the combined use of essential oils with the same carrier oils as carried out in this study [31]. The synergy was consistent for several of the compounds and essential oils across the two studies. This could be observed for the synergistic combination of *p*-cymene with *A. vera.* At 24 h, the essential oils *K. ericoides* and *Melaleuca alternifolia* Cheel (tea tree), containing the compound *p*-cymene (11.9% and 9.6%, respectively), showed synergy when combined with *A. vera* [31]. This could suggest a correlation between the synergistic activity seen with the essential oil: carrier oil combinations and the synergistic activity seen with the essential oil compound: carrier oil combinations.

The previous study also found the carrier oils *A. vera* and *S. chinensis* to reduce the toxicity of the essential oils at 24 h and *A. vera* and *P. armeniaca* to cause the most reduction in toxicity at 48 h. *Aloe vera* was present in most synergistic essential oil–carrier oil combinations over 24 and 48 h [31]. Some differences in the results between this study and the previous one suggests that the toxicity patterns shown by the combined use of carrier oils and essential oils cannot always be generalized to predict which carrier oil would be most advantageous in decreasing the toxicity of the compounds. The essential oil *L. angustifolia*, containing linalyl acetate (35.6%), linalool (32.8%), and β-caryophyllene (10.2%) as its major compounds, resulted in synergy when combined with *C. officinalis* [31], and in this study, linalyl acetate resulted in synergy with *C. officinalis* whereas β-caryophyllene and linalool did not. This observed difference may also be due to the mixture of compounds in the neat essential oil reacting differently when compared to examining combinations with single compounds.

### 2.3. Selectivity Index

The selectivity index for all the combinations which showed antimicrobial synergy was calculated (Table 18). Various interpretations exist, however, this study considers a selectivity index of >4 as being acceptable, when the antimicrobial benefit is not lost due to the toxicity [71]. A selectivity index below four indicates that the toxicity of the compound: carrier oil combination is too high and the antimicrobial activity is most likely attributed to the toxicity of the sample and not the interaction [71]. Of the 23 synergistic combinations, 10 at 24 h and 9 at 48 h had SI values of >4, with thymoquinone being the main compound present in these combinations.

## 3. Materials and Methods

### 3.1. Sample Selection and Preparation

A selection of 21 essential oil compounds (Sigma-Aldrich, Johannesburg, South Africa) were selected based on their previously reported noteworthy antimicrobial activity [2,4,31,72,73,74,75,76]. All carrier oils were obtained from Escentia (Johannesburg, South Africa) and Scatters Oils (Johannesburg, South Africa) and consisted of *Aloe vera* (Aloe vera); *Calendula officinalis* (Calendula); *Hypericum perforatum* (St John’s wort); *Persea americana* (Avocado); *Prunus armeniaca* (Apricot kernel); and *Simmondsia chinensis* (Jojoba). The carrier oil selection was based on their frequent use in aromatherapy and relevance to application on the skin.

### 3.2. Culture Preparation

The micro-organisms selected for this study included the ESKAPE pathogens and one yeast pathogen. Selection was based on their importance in contributing towards antimicrobial resistance [2,77]. The investigated bacteria included *Enterococcus faecium* (ATCC 27270), *Staphylococcus aureus* (ATCC 25923), *Klebsiella pneumoniae* (ATCC 13883), *Acinetobacter baumannii* (ATCC 17606), *Pseudomonas aeruginosa* (ATCC 27858), and *Escherichia coli* (ATCC 8739). The pathogen reference strain *Candida albicans* (ATCC 10231) was selected as a yeast representative. The micro-organisms were cultured in Tryptone Soya broth (TSB) (Oxoid), and Tryptone Soya agar (TSA) and were incubated at 37 °C for 24 h (bacteria) and at 37 °C for 48 h (yeast). The purity of the micro-organisms was confirmed by streaking each culture onto an agar plate and ensuring growth of single colonies, as well as checking colony morphology with visual standards within the microbiology laboratory.

### 3.3. Sample Preparation

For the broth microdilution assay, the samples were diluted to a concentration of 32.00 mg/mL in acetone. For the brine shrimp lethality assay, all selected samples were prepared in 2% dimethyl sulfoxide (DMSO) or 20–50% acetone at a concentration of 2.00 mg/mL depending on solubility.

### 3.4. Antimicrobial Analysis

The broth microdilution method using a 96-well microtiter plate, as described in a previous study [5], was used to quantify the inhibitory activity of the compounds and carrier oils. Preparation of the microtiter plates involved the aseptic addition of 100.00 µL of TSB into each of the wells of the microtiter plate. The samples were then added, at a volume of 100.00 µL, to the first row of the plate. When testing the combinations, a modification was made where 50.00 µL of the compound and 50.00 µL of the carrier oil were placed in the first row of wells (to make up 100 µL of sample) of the plate. A volume of 100.00 µL of a positive, negative, and culture control were included for each strain studied. The positive control (0.01 mg/mL ciprofloxacin for bacteria or 0.1 mg/mL nystatin for yeast) was used to ensure microbial susceptibility. The negative control (32.00 mg/mL water in acetone) was included to rule out whether the antimicrobial activity was attributed to the solvent. A culture control in TSB was included to ensure the broth supported growth of the reference strains. The samples were then serially diluted down the rows in concentrations of 8.00; 4.00; 2.00; 1.00; 0.50; 0.25; 0.13; and 0.06 mg/mL. After the preparation of an approximate inoculum concentration of 1 × 10⁶ colony-forming units (CFU)/mL for each reference strain, 100.00 µL was added to each of the wells. A sterile adhesive sealing film was used to seal the microtiter plate to prevent loss of the samples through evaporation. Incubation of the microtiter plates occurred at 37 °C for 24 h for bacteria and 37 °C for 48 h for the yeast. A volume of 40.00 µL of *p*-iodonitrotetrazolium violet solution (INT) (Sigma-Aldrich), at a concentration of 0.04 mg/mL, was then added to each well after incubation. The lowest concentration with no colour change was taken as the minimum inhibitory concentration (MIC) for that sample. All samples were tested in triplicate. The average of the samples was calculated and the standard deviation (SD) determined using Microsoft Excel (Microsoft Office Home and Student 2016). Results were considered noteworthy if the MIC value was ≤1.00 mg/mL [5].

### 3.5. Toxicity Studies

The brine shrimp lethality assay [78] was used to determine the toxicity of 21 compounds and six carrier oils alone and in combination. Artificial seawater was prepared by dissolving 16.00 g of Tropic Marine^®^ sea salt in 500.00 mL of distilled water. This solution was transferred into a bottomless, inverted receptacle. Dried brine shrimp (*Artemia franciscana*) eggs, from Ocean Nutrition^TM^, were added to the salt water. Aeration of the water with a rotary pump was included to ensure a high brine shrimp hatch rate. A constant source of light and warmth, from a 220 to 240 V lamp, was used to assist with the hatching process. The eggs were incubated at 25 °C for 24–48 h. For the assay, a 48-well microtiter plate was prepared by adding 400.00 µL of salt water containing 40–60 live brine shrimp to each well. A volume of 400.00 µL of sample was added to each well. For the combinations, a 1:1 ratio of 200 µL each of each sample (carrier oil: compound) was prepared prior to being added to the well containing the shrimp. The assay included a negative, non-toxic control of 32.00 g/L of artificial seawater to ensure the promotion of growth and survival of the brine shrimp. The positive control in the assay consisted of 1.60 mg/mL of potassium dichromate, a highly toxic compound. At 0, 24 and 48 h, the dead brine shrimp were viewed and counted under a light microscope (Olympus) at 40× magnification. A lethal dose of acetic acid (Saarchem; 100% (*v/v*); 50.00 μL) was added to each well and a final count of dead brine shrimp taken [79]. Then, the percentage mortality was calculated using Equation (1). Biological toxicity was considered for a percentage mortality of 50% or greater [80]. All studies were carried out in triplicate. The average percentage mortality of the brine shrimp was recorded on Microsoft Excel (Microsoft Office Home and Student 2016).
(1)$$\%\;\mathrm{Mortality}=\frac{\mathrm{Dead}\;\mathrm{shrimp}\;\mathrm{at}\frac{24}{48}h\;\left(\mathrm{before}\;\mathrm{acetic}\;\mathrm{acid}\right)-\mathrm{Dead}\;\mathrm{shrimp}\;\left(\mathrm{time}=0\right)}{\mathrm{Dead}\;\mathrm{shrimp}\;\left(\mathrm{after}\;\mathrm{acetic}\;\mathrm{acid}\right)\;\times \;100}$$

### 3.6. Interactive Profiles of Combinations

The interactive profiles of the combinations for the antimicrobial and toxicity assays were undertaken, and the fractional inhibitory concentration index (ΣFIC antimicrobial) or the fractional percentage mortality (ΣFPM toxicity) was calculated, respectively, according to Equation (2).

The (a*) represents the essential oil compound in combination and (b*) represents the carrier oil.
(2)$$\begin{array}{c} \mathrm{FIC}\;\mathrm{or}\;\mathrm{FPM}\;(i)=\frac{\;\left(a*\right)\mathrm{combined}\;\mathrm{with}\;\left(b*\right)}{\left(a\right)\;\mathrm{independently}\;}\\ \mathrm{FIC}\;\mathrm{or}\;\mathrm{FPM}\;(\mathrm{ii})=\frac{\;\left(b\right)\;\mathrm{combined}\;\mathrm{with}\;\left(a\right)}{\;\left(b\right)\;\mathrm{independently}\;}\\ \Sigma \mathrm{FIC}\;=\;\mathrm{FIC}\;\left(i\right)+\;\mathrm{FIC}\;\left(\mathrm{ii}\right)\;\mathrm{or}\;\Sigma \mathrm{FPM}\;=\;\mathrm{FPM}\;\left(i\right)+\;\mathrm{FPM}\;\left(\mathrm{ii}\right)\; \end{array}$$

The interactive profile was interpreted as follows: an ΣFIC or ΣFPM value of ≤0.5 represented synergy, >0.5–1.0 indicated additive interactions, >1.0–≤4.0 demonstrated indifference, and a value > 4.0 indicated antagonism [81]. For antimicrobial studies, a synergistic combination is regarded as having increased antimicrobial activity and an antagonistic combination is regarded as having decreased antimicrobial activity. Where MIC values of >8.00 mg/mL were determined, they were recorded as 16.00 mg/mL for the purpose of calculating an ƩFIC value. For toxicity studies, synergy is due to a decrease in toxicity of the compounds.

### 3.7. Selectivity Index (SI)

The selectivity index indicates the ratio of toxicity to antimicrobial activity of a sample and was calculated using Equation (3).
(3)$$\mathrm{SI}=\frac{{\mathrm{LC}}_{50}}{\mathrm{MIC}}$$

## 4. Conclusions

This study investigated the antimicrobial and toxicity effects of carrier oils in combination with essential oil compounds. When looking at the antimicrobial activity of 882 combinations, 3% of combinations were synergistic and 4% were antagonistic. The compound thymoquinone and the carrier oil *P. armeniaca* were present in the majority of the antimicrobial synergistic combinations, and the compound santalol and carrier oil *H. perforatum* were found in the majority of the antagonistic combinations.

When investigating the toxicity interactions of 105 combinations at 24 h, 10% of the combinations were synergistic, and 77% were antagonistic. When investigating the toxicity of 126 combinations at 48 h, 6% of the combinations were synergistic and 71% were antagonistic. These antagonistic interactions warrant caution when combining equal ratios of compound to carrier oil. The carrier oil *C. officinalis* was present in the majority of the antagonistic toxicity combinations at 24 h, and the carrier oil *P. armeniaca* was present in the majority of the synergistic toxicity combinations at 48 h. The selectivity index demonstrated thymoquinone to be the most favorable compound in combination with carrier oils because it was present in the majority of combinations that had an SI value of >4.

Future studies investigating varying ratios may provide a more optimal toxicity profile. It may also be beneficial to investigate the various constituents of the carrier oils themselves, such as the separate free fatty acids and the vitamins, to determine their influence on the essential oil compound toxicity and antimicrobial activity. Nonetheless, this study provides valuable insight into the antimicrobial and toxicity effects of carrier oils when combined with essential oil compounds.

## Figures and Tables

**Figure 1 molecules-28-00030-f001:**
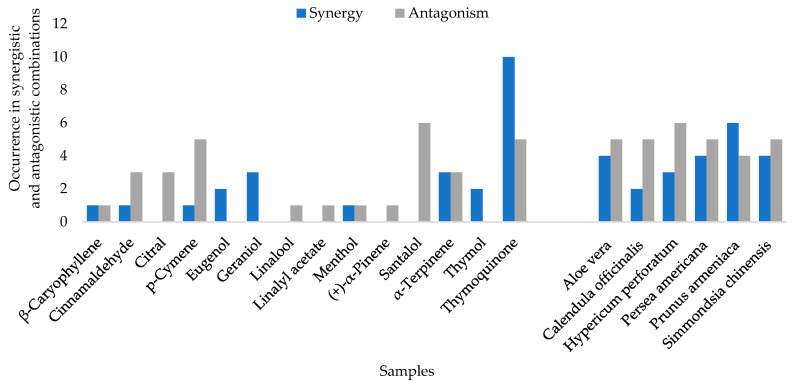
Summary of compounds and carrier oils occurring in synergistic and antagonistic interactions.

**Table 1 molecules-28-00030-t001:** Antimicrobial activity (MIC in mg/mL with standard deviation (SD) in parentheses) of carrier oils (*n* = 3) against all pathogens.

Carrier Oil	Micro-Organism						
	Gram-Positive Bacteria		Gram-Negative Bacteria				Yeast
	*E. faecium* ATCC 27270	*S. aureus* ATCC 25923	*K. pneumoniae* ATCC 13883	*A. baumannii* ATCC 17606	*P. aeruginosa* ATCC 27853	*E. coli* ATCC 8729	*C. albicans* ATCC 10231
*Aloe vera* Mill.	16.00 (±0.00)	2.00 (±0.00)	2.00 (±0.00)	3.43 (±0.90)	1.75 (±0.43)	3.00 (±1.00)	**1.00 (±0.00) ^1^**
*Calendula officinalis* L.	16.00 (±0.00)	2.00 (±0.00)	2.00 (±0.00)	3.60 (±0.80)	2.25 (±1.09)	2.50 (±0.87)	**1.00 (±0.00)**
*Hypericum perforatum* L.	16.00 (±0.00)	2.67 (±0.94)	2.00 (±0.00)	3.60 (±0.80)	1.75 (±0.43)	2.50 (±0.87)	1.50 (±0.50)
*Persea americana* Mill.	4.00 (±0.00)	2.50 (±0.87)	2.00 (±0.00)	3.00 (±1.00)	2.50 (±0.87)	2.50 (±0.87)	3.00 (±1.00)
*Prunus armeniaca* Blanco.	4.00 (±0.00)	2.50 (±0.87)	2.00 (±0.00)	2.00 (±0.00)	2.67 (±0.94)	3.50 (±0.87)	2.00 (±0.00)
*Simmondsia chinensis* C.K. Schneid.	4.00 (±0.00)	3.00 (±1.00)	2.00 (±0.00)	4.00 (±0.00)	3.33 (±0.94)	3.50 (±0.87)	**1.00 (±0.00)**
Positive control ^2^	1.56 ^2.1^	0.73 ^2.1^	0.20 ^2.1^	0.52 ^2.1^	1.25 ^2.1^	0.50 ^2.1^	0.94 ^2.2^
Negative control	> 8.00	3.30	> 8.00	3.00	>8.00	> 8.00	2.00
Culture control	>8.00	>8.00	>8.00	>8.00	>8.00	>8.00	>8.00

^1^ Noteworthy MIC values in bold; ^2^ Ciprofloxacin ^2.1^ /Nystatin ^2.2^ (µg/mL).

**Table 2 molecules-28-00030-t002:** Antimicrobial activity (MIC in mg/mL and SD in parentheses) of the essential oil compounds (*n* = 3).

Essential Oil Compound	Micro-Organism						
	Gram-Positive Bacteria		Gram-Negative Bacteria				Yeast
	*E. faecium* ATCC 27270	*S. aureus* ATCC 25923	*K. pneumoniae* ATCC 13883	*A. baumannii* ATCC 17606	*P. aeruginosa* ATCC 27853	*E. coli* ATCC 8729	*C. albicans* ATCC 10231
Carvacrol	**0.67 (±0.24)** ^1^	**0.42 (±0.12)**	**0.25 (±0.00)**	**0.83 (±0.24)**	**0.50 (±0.18)**	**0.63 (±0.22)**	**0.25 (±0.00)**
β-Caryophyllene	4.00 (±0.00)	3.00 (±1.00)	2.00 (±0.00)	3.33 (±0.94)	2.67 (±0.94)	3.33 (±0.94)	2.00 (±0.00)
Cinnamaldehyde	**0.50 (±0.00)**	**0.13 (±0.00)**	**0.33 (±0.12)**	**0.42 (±0.12)**	**0.38 (±0.13)**	**0.38 (±0.13)**	**0.03 (±0.00)**
Citral	1.33 (±0.47)	**0.25 (±0.72)**	**1.00 (±0.00)**	**1.00 (±0.00)**	1.75 (±0.43)	**1.00 (±0.00)**	**0.50 (±0.00)**
*p*-Cymene	3.33 (±0.94)	2.50 (±0.87)	**1.00 (±0.00)**	3.33 (±0.94)	3.00 (±1.00)	3.00 (±1.00)	1.50 (±0.50)
Eugenol	**1.00 (±0.00)**	1.25 (±0.43)	**0.50 (±0.00)**	**1.00 (±0.00)**	**1.00 (±0.00)**	**1.00 (±0.00)**	**0.38 (±0.13)**
Geraniol	1.67 (±0.47)	1.25 (±0.43)	**0.83 (±0.24)**	**1.00 (±0.00)**	1.25 (±0.43)	1.20 (±0.40)	**0.25 (±0.00)**
Isoeugenol	**0.83 (±0.24)**	**0.33 (±0.12)**	**0.63 (±0.22)**	**0.58 (±0.19)**	**0.55 (±0.24)**	**1.00 (±0.00)**	**0.25 (±0.00)**
R(+)-Limonene	3.00 (±1.00)	3.33 (±0.94)	3.00 (±1.00)	4.00 (±0.00)	2.25 (±1.09)	5.33 (±1.89)	**1.00 (±0.00)**
Linalool	3.33 (±0.94)	1.40 (±1.07)	**1.00 (±0.00)**	2.00 (±0.00)	1.75 (±0.43)	3.00 (±1.00)	**1.00 (±0.00)**
Linalyl acetate	4.00 (±0.00)	2.50 (±0.87)	1.67 (±0.47)	2.00 (±0.00)	1.75 (±0.43)	3.00 (±1.00)	**1.00 (±0.00)**
Menthol	2.67 (±0.94)	**1.00 (±0.00)**	1.67 (±0.47)	2.00 (±0.00)	2.00 (±0.00)	2.00 (±0.00)	**0.50 (±0.00)**
Nerol	**1.00 (±0.00)**	1.33 (±0.47)	1.50 (±0.50)	1.50 (±0.50)	1.33 (±0.47)	1.67 (±0.47)	**0.50 (±0.00)**
(+)-α-Pinene	3.00 (±1.00)	2.00 (±0.00)	3.00 (±1.00)	3.00 (±1.00)	2.00 (±0.00)	6.00 (±2.00)	**1.00 (±0.00)**
Santalol	**0.25 (±0.00)**	**0.19 (±0.06)**	4.00 (±0.00)	**1.00 (±0.00)**	4.00 (±0.00)	**0.50 (±0.00)**	**1.00 (±0.00)**
α-Terpinene	7.00 (±1.73)	2.00 (±0.00)	3.00 (±1.00)	2.00 (±0.00)	2.67 (±0.94)	3.33 (±0.94)	**1.00 (±0.00)**
γ-Terpinene	3.33 (±0.94)	2.67 (±0.94)	3.67 (±3.09)	4.00 (±2.83)	4.00 (±2.45)	4.00 (±0.00)	2.00 (±0.00)
(+)-Terpinen-4-ol	4.00 (±0.00)	3.00 (±1.00)	1.50 (±0.50)	3.00 (±1.00)	2.00 (±0.00)	2.50 (±0.87)	2.00 (±0.00)
α-Terpineol	2.00 (±0.00)	1.33 (±0.47)	**1.00 (±0.00)**	2.00 (±0.00)	2.00 (±0.00)	2.00 (±0.00)	**0.75 (±0.25)**
Thymol	**0.67 (±0.24)**	**0.75 (±0.25)**	**0.25 (±0.00)**	**1.00 (±0.00)**	**0.25 (±0.00)**	**0.75 (±0.25)**	**0.50 (±0.00)**
Thymoquinone	**0.01 (±0.00)**	**0.00 (±0.00)**	**1.00 (±0.00)**	**0.02 (±0.01)**	1.50 (±0.50)	**0.13 (±0.00)**	**0.08 (±0.05)**
Positive control ^2^	1.09 ^2.1^	0.50 ^2.1^	0.08 ^2.1^	0.66 ^2.1^	0.36 ^2.1^	0.88 ^2.1^	1.25 ^2.2^
Negative control (water in acetone)	>8.00	3.33	4.67	>8.00	4	5	8
Culture control	>8.00	>8.00	>8.00	>8.00	>8.00	>8.00	>8.00

^1^ Noteworthy MIC values in bold; ^2^ Ciprofloxacin ^2.1^ /Nystatin ^2.2^ (µg/mL).

**Table 3 molecules-28-00030-t003:** Antimicrobial activity (MIC in mg/mL and ΣFIC) of the essential oil compound: carrier oil combinations against *Enterococcus faecium* ATCC 27270 (*n* = 3).

Compounds	Carrier Oils											
	*A. vera*		*C. officinalis*		*H. perforatum*		*P. americana*		*P. armeniaca*		*S. chinensis*	
	MIC	ΣFIC	MIC	ΣFIC	MIC	ΣFIC	MIC	ΣFIC	MIC	ΣFIC	MIC	ΣFIC
Carvacrol	1.50	1.17	1.50	1.17	**1.00**	0.78	**1.00**	0.88	**1.00**	0.88	**1.00**	0.88
β-Caryophyllene	6.00	0.94	4.00	0.63	6.00	0.94	8.00	2.00	4.00	1.00	4.00	1.00
Cinnamaldehyde	**1.00** ^1^	1.03	**1.00**	1.03	**0.83**	0.86	**1.00**	1.13	**1.00**	1.13	**1.00**	1.13
Citral	2.00	0.81	3.00	1.22	2.00	0.81	2.00	1.00	2.00	1.00	2.00	1.00
*p*-Cymene	16.00	2.90	16.00	2.90	7.00	1.27	4.00	1.10	16.00	*4.40* ^3^	4.00	1.10
Eugenol	4.00	2.13	2.00	1.06	2.00	1.06	2.00	1.25	2.00	1.25	2.00	1.25
Geraniol	2.00	0.66	2.00	0.66	2.00	0.66	2.00	0.85	2.00	0.85	2.00	0.85
Isoeugenol	2.00	1.26	2.00	1.26	1.50	0.95	1.50	1.09	2.00	1.45	**1.00**	0.73
R(+)-Limonene	4.00	0.79	4.00	0.79	4.00	0.79	4.00	1.17	4.00	1.17	4.00	1.17
Linalool	4.00	0.73	4.00	0.73	4.00	0.73	4.00	1.10	4.00	1.10	4.00	1.10
Linalyl acetate	16.00	2.50	4.00	0.63	4.00	0.63	4.00	1.00	4.00	1.00	4.00	1.00
Menthol	2.00	** *0.44* ** ^2^	8.00	1.75	4.00	0.88	3.00	0.94	4.00	1.25	2.00	0.63
Nerol	2.00	1.06	2.00	1.06	3.00	1.59	3.00	1.88	2.00	1.25	2.00	1.25
(+)-α-Pinene	4.00	0.79	4.00	0.79	4.00	0.79	4.00	1.17	4.00	1.17	4.00	1.17
Santalol	1.50	3.05	1.50	3.05	**0.50**	1.02	**0.50**	1.06	**0.50**	1.06	**0.50**	1.06
α-Terpinene	4.00	** *0.41* **	4.00	** *0.41* **	4.00	** *0.41* **	8.00	1.57	16.00	3.14	4.00	0.79
γ-Terpinene	16.00	2.90	4.00	0.73	6.00	1.09	4.00	1.10	4.00	1.10	4.00	1.10
(+)-Terpinen-4-ol	4.00	0.63	4.00	0.63	4.00	0.63	4.00	1.00	4.00	1.00	4.00	1.00
α-Terpineol	4.00	1.13	4.00	1.13	4.00	1.13	4.00	1.50	4.00	1.50	4.00	1.50
Thymol	2.00	1.56	2.00	1.56	2.00	1.56	3.00	2.63	2.00	1.75	2.00	1.75
Thymoquinone	**0.08**	*5.00*	**0.13**	*8.00*	**0.19**	*12.01*	**0.13**	*8.02*	**0.03**	2.00	**0.13**	*8.02*
Positive control ^3^	1.15		1.25		1.09		1.25		1.15		2.50	
Negative control	>8.00		>8.00		>8.00		>8.00		>8.00		>8.00	
Culture control	>8.00		>8.00		>8.00		>8.00		>8.00		>8.00	

^1^ Noteworthy MIC values in bold; ^2^ synergy in bold and italics; ^3^ antagonism in italics.

**Table 4 molecules-28-00030-t004:** Antimicrobial activity (MIC in mg/mL and ΣFIC) of the essential oil compound: carrier oil combinations against *Staphylococcus aureus* ATCC 25923 (*n* = 3).

Compounds	Carrier Oils											
	*A. vera*		*C. officinalis*		*H. perforatum*		*P. americana*		*P. armeniaca*		*S. chinensis*	
	MIC	ΣFIC	MIC	ΣFIC	MIC	ΣFIC	MIC	ΣFIC	MIC	ΣFIC	MIC	ΣFIC
Carvacrol	1.33	1.93	1.00	1.25	1.33	1.85	0.38	0.53	0.50	0.70	0.50	0.68
β-Caryophyllene	1.67	0.69	1.33	0.56	4.00	1.42	2.00	0.73	3.00	1.10	2.00	0.67
Cinnamaldehyde	1.33	*5.67* ^3^	**1.00**	*4.25*	**1.00**	*4.19*	**0.25**	1.05	**0.25**	1.05	**0.13**	0.52
Citral	3.00	*6.75*	3.33	*7.50*	2.00	*4.37*	**0.50**	1.10	**0.50**	1.10	**0.38**	0.81
*p*-Cymene	2.00	0.90	8.00	3.60	16.00	*6.20*	2.67	1.07	3.33	1.33	2.67	0.98
Eugenol	2.00	1.30	2.00	1.30	3.33	1.96	**1.00**	0.60	**0.50**	** *0.30* **	**0.50**	** *0.28* **
Geraniol	**1.00** ^1^	0.65	1.33	0.86	2.00	1.17	**0.75**	** *0.45* **	**0.75**	** *0.45* **	**0.75**	** *0.43* **
Isoeugenol	2.00	3.50	**1.00**	1.75	2.00	3.37	**0.50**	0.85	**0.50**	0.85	**0.50**	0.83
R(+)-Limonene	2.67	1.07	2.67	1.07	2.67	0.90	2.00	0.70	2.67	0.93	4.00	1.27
Linalool	4.00	2.43	4.00	2.43	4.00	2.18	1.50	0.84	2.67	1.49	2.67	1.40
Linalyl acetate	5.33	2.40	4.00	1.80	3.00	1.16	3.33	1.33	2.00	0.80	2.67	0.98
Menthol	4.00	3.00	2.00	1.50	3.00	2.06	**1.00**	0.70	**1.00**	0.70	2.00	1.33
Nerol	1.50	0.94	2.67	1.67	2.67	1.50	2.67	1.53	1.50	0.86	2.67	1.44
(+)-α-Pinene	2.00	1.00	2.00	1.00	3.00	1.31	2.00	0.90	2.00	0.90	2.00	0.83
Santalol	3.00	*8.75*	2.00	*5.83*	**1.00**	2.85	**0.75**	2.15	**1.00**	2.87	**0.50**	1.42
α-Terpinene	4.00	2.00	3.33	1.67	16.00	*7.00*	3.00	1.35	2.00	0.90	2.00	0.83
γ-Terpinene	2.00	0.88	4.00	1.75	8.00	3.00	2.00	0.78	2.00	0.78	3.00	1.06
(+)-Terpinen-4-ol	2.00	0.83	2.00	0.83	2.00	0.71	3.00	1.10	3.00	1.10	2.00	0.67
α-Terpineol	4.00	2.50	4.00	2.50	4.00	2.25	1.50	0.86	**1.00**	0.58	2.00	1.08
Thymol	**1.00**	0.92	**0.75**	0.69	**0.75**	0.64	**0.50**	** *0.43* **	**0.50**	** *0.43* **	**0.75**	0.63
Thymoquinone	**0.001**	** *0.19* ** ^2^	**0.003**	** *0.38* **	**0.001**	** *0.13* **	**0.003**	** *0.38* **	**0.001**	** *0.19* **	**0.002**	** *0.25* **
Positive control	1.88		1.88		1.88		1.46		1.46		1.46	
Negative control	4.00		>8.00		>8.00		>8.00		>8.00		>8.00	
Culture control	>8.00		>8.00		>8.00		>8.00		>8.00		>8.00	

^1^ Noteworthy MIC values in bold; ^2^ synergy in bold and italics; ^3^ antagonism in italics.

**Table 5 molecules-28-00030-t005:** Antimicrobial activity (MIC in mg/mL and ΣFIC) of the essential oil compound: carrier oil combinations against *Klebsiella pneumoniae* ATCC 13883 (*n* = 3).

Compounds	Carrier Oils											
	*A. vera*		*C. officinalis*		*H. perforatum*		*P. americana*		*P. armeniaca*		*S. chinensis*	
	MIC	ΣFIC	MIC	ΣFIC	MIC	ΣFIC	MIC	ΣFIC	MIC	ΣFIC	MIC	ΣFIC
Carvacrol	**1.00** ^1^	2.25	**0.50**	1.13	**0.75**	1.69	0.75	1.69	**1.00**	2.25	**0.38**	0.84
β-Caryophyllene	2.50	1.25	3.00	1.50	2.00	1.00	2.00	1.00	2.00	1.00	2.33	1.17
Cinnamaldehyde	**0.50**	0.88	**0.50**	0.88	**0.50**	0.88	0.50	0.88	**0.50**	0.88	**0.50**	0.88
Citral	2.00	1.50	2.00	1.50	2.00	1.50	3.00	2.25	2.00	1.50	2.00	1.50
*p*-Cymene	2.00	1.50	2.00	1.50	2.00	1.50	3.00	2.25	2.00	1.50	2.00	1.50
Eugenol	**1.00**	1.25	**1.00**	1.25	1.50	1.88	**1.00**	1.25	**1.00**	1.25	**1.00**	1.25
Geraniol	**1.00**	0.85	1.50	1.28	1.50	1.28	**1.00**	0.85	**1.00**	0.85	**1.00**	0.85
Isoeugenol	**1.00**	1.05	**1.00**	1.05	**1.00**	1.05	**1.00**	1.05	**1.00**	1.05	**1.00**	1.05
R(+)-Limonene	2.00	0.83	2.00	0.83	2.00	0.83	2.00	0.83	2.00	0.83	2.00	0.83
Linalool	3.00	2.25	1.50	1.13	2.00	1.50	2.00	1.50	2.00	1.50	2.00	1.50
Linalyl acetate	2.00	1.10	2.00	1.10	2.00	1.10	2.00	1.10	2.00	1.10	3.00	1.65
Menthol	2.00	1.10	1.50	0.83	2.00	1.10	2.00	1.10	1.50	0.83	1.50	0.83
Nerol	2.00	1.17	2.00	1.17	3.00	1.75	2.00	1.17	2.00	1.17	**1.00**	0.58
(+)-α-Pinene	2.00	0.83	3.00	1.25	1.50	0.63	2.00	0.83	2.00	0.83	2.00	0.83
Santalol	3.00	1.13	3.00	1.13	2.00	0.75	16.00	*6.00* ^2^	2.00	0.75	2.00	0.75
α-Terpinene	2.00	0.83	2.00	0.83	2.00	0.83	2.00	0.83	2.00	0.83	2.00	0.83
γ-Terpinene	2.00	0.77	2.00	0.77	2.00	0.77	2.00	0.77	2.00	0.77	1.50	0.58
(+)-Terpinen-4-ol	2.00	1.17	3.00	1.75	1.50	0.88	2.00	1.17	2.00	1.17	1.50	0.88
α-Terpineol	**1.00**	0.75	**1.00**	0.75	2.00	1.50	2.00	1.50	2.00	1.50	**1.00**	0.75
Thymol	**1.00**	2.25	**1.00**	2.25	**1.00**	2.25	**1.00**	2.25	**0.75**	1.69	**1.00**	2.25
Thymoquinone	**1.00**	0.75	**1.00**	0.75	2.00	1.50	2.00	1.50	4.00	3.00	2.00	1.50
Positive control	0.08		0.08		0.10		0.10		0.10		0.08	
Negative control	>8.00		>8.00		>8.00		>8.00		>8.00		>8.00	
Culture control	>8.00		>8.00		>8.00		>8.00		>8.00		>8.00	

^1^ Noteworthy MIC values are shown in bold; ^2^ antagonism in italics.

**Table 6 molecules-28-00030-t006:** Antimicrobial activity (MIC in mg/mL and ΣFIC) of the essential oil compound: carrier oil combinations against *Acinetobacter baumannii* ATCC 17606 (*n* = 3).

Compounds	Carrier Oils											
	*A. vera*		*C. officinalis*		*H. perforatum*		*P. americana*		*P. armeniaca*		*S. chinensis*	
	MIC	ΣFIC	MIC	ΣFIC	MIC	ΣFIC	MIC	ΣFIC	MIC	ΣFIC	MIC	ΣFIC
Carvacrol	**1.00** ^1^	0.75	**1.00**	0.74	2.00	1.48	**1.00**	0.77	1.50	1.28	**1.00**	0.73
β-Caryophyllene	4.00	1.18	16.00	*4.62* ^2^	4.00	1.16	6.67	2.11	4.00	1.60	4.00	1.10
Cinnamaldehyde	**0.50**	0.67	**0.75**	1.00	**0.75**	1.00	**1.00**	1.37	**0.75**	1.09	**0.75**	0.99
Citral	2.00	1.29	**1.00**	0.64	1.50	0.96	1.50	1.00	1.50	1.13	2.00	1.25
*p*-Cymene	4.00	1.18	4.00	1.16	4.00	1.16	16.00	*5.07*	4.00	1.60	4.00	1.10
Eugenol	2.00	1.29	2.00	1.28	2.00	1.28	2.00	1.33	2.00	1.50	2.00	1.25
Geraniol	2.00	1.29	1.50	0.96	2.00	1.28	2.00	1.33	2.00	1.50	2.00	1.25
Isoeugenol	1.50	1.50	2.00	1.99	2.00	1.99	**1.00**	1.02	1.50	1.66	**1.00**	0.98
R(+)-Limonene	4.00	1.08	4.00	1.06	4.00	1.06	4.00	1.17	8.00	3.00	4.00	1.00
Linalool	3.00	1.19	2.00	0.78	4.80	1.87	2.00	0.83	4.00	2.00	6.00	2.25
Linalyl acetate	4.00	1.58	3.00	1.17	4.00	1.56	4.00	1.67	4.00	2.00	4.00	1.50
Menthol	4.00	1.58	2.00	0.78	4.00	1.56	2.00	0.83	3.33	1.67	4.00	1.50
Nerol	2.00	0.96	3.00	1.42	3.00	1.42	4.00	2.00	3.00	1.75	2.00	0.92
(+)-α-Pinene	3.00	0.94	4.00	1.22	4.00	1.22	4.00	1.33	16.00	*6.67*	4.00	1.17
Santalol	**1.00**	0.65	3.00	1.92	4.00	2.56	4.00	2.67	6.00	*4.50*	3.00	1.88
α-Terpinene	4.00	1.58	4.00	1.56	3.00	1.17	16.00	*6.67*	4.00	2.00	4.00	1.50
γ-Terpinene	5.33	1.44	3.33	0.88	4.00	1.06	2.00	0.58	4.00	1.50	4.00	1.00
(+)-Terpinen-4-ol	4.00	1.25	4.00	1.22	4.00	1.22	4.00	1.33	4.00	1.67	4.00	1.17
α-Terpineol	3.00	1.19	4.00	1.56	2.00	0.78	3.33	1.39	3.00	1.50	4.00	1.50
Thymol	2.00	1.29	2.00	1.28	2.00	1.28	2.00	1.33	2.00	1.50	2.00	1.25
Thymoquinone	**0.05**	1.01	**0.03**	0.67	**0.05**	1.01	**0.03**	0.67	**0.03**	0.67	**0.03**	0.67
Positive control	0.53		0.43		0.72		0.72		0.63		0.63	
Negative control	>8.00		>8.00		>8.00		>8.00		>8.00		>8.00	
Culture control	>8.00		>8.00		>8.00		>8.00		>8.00		>8.00	

^1^ Noteworthy MIC values in bold; ^2^ antagonism in italics.

**Table 7 molecules-28-00030-t007:** Antimicrobial activity (MIC in mg/mL and ΣFIC) of the essential oil compound: carrier oil combinations against *Pseudomonas aeruginosa* ATCC 27853 (*n* = 3).

Compounds	Carrier Oils											
	*A. vera*		*C. officinalis*		*H. perforatum*		*P. americana*		*P. armeniaca*		*S. chinensis*	
	MIC	ΣFIC	MIC	ΣFIC	MIC	ΣFIC	MIC	ΣFIC	MIC	ΣFIC	MIC	ΣFIC
Carvacrol	**0.67** ^1^	0.86	**1.00**	1.22	**0.50**	0.64	**0.67**	0.80	**0.67**	0.79	**0.67**	0.77
β-Caryophyllene	2.00	0.95	3.00	1.23	2.00	0.95	2.00	0.78	3.00	1.12	3.00	1.01
Cinnamaldehyde	**0.50**	0.81	**0.50**	0.78	**0.67**	1.08	**0.50**	0.77	**0.50**	0.76	**0.50**	0.74
Citral	1.67	0.95	3.00	1.52	3.00	1.71	1.67	0.81	1.67	0.79	1.67	0.73
*p*-Cymene	16.00	*7.24* ^2^	2.67	1.04	2.00	0.90	1.67	0.61	1.67	0.59	4.00	1.27
Eugenol	1.33	1.05	1.33	0.96	1.33	1.05	1.33	0.93	1.33	0.92	1.33	0.87
Geraniol	1.33	0.91	1.67	1.04	1.67	1.14	1.33	0.80	1.83	1.08	1.67	0.92
Isoeugenol	**1.00**	1.19	**0.83**	0.94	**0.83**	1.00	**0.83**	0.92	**0.83**	0.91	**1.00**	1.06
R(+)-Limonene	6.00	3.05	2.67	1.19	1.67	0.85	2.00	0.84	3.00	1.23	1.67	0.62
Linalool	1.67	0.95	4.00	2.03	3.00	1.71	1.67	0.81	3.00	1.42	1.67	0.73
Linalyl acetate	1.67	0.95	2.50	1.27	3.33	1.90	2.00	0.97	1.67	0.79	3.33	1.45
Menthol	3.00	1.61	3.00	1.42	4.00	2.14	2.00	0.90	3.00	1.31	2.00	0.80
Nerol	1.67	1.10	1.67	1.00	1.67	1.10	1.67	0.96	1.67	0.94	1.67	0.88
(+)-α-Pinene	2.00	1.07	2.00	0.94	2.00	1.07	2.00	0.90	2.00	0.87	2.00	0.80
Santalol	4.00	1.64	2.00	0.69	2.00	0.82	3.00	0.98	2.00	0.62	2.00	0.55
α-Terpinene	4.00	1.89	2.00	0.82	2.00	0.95	5.33	2.07	2.00	0.75	3.00	1.01
γ-Terpinene	3.00	1.23	2.00	0.69	8.00	3.29	3.00	0.98	3.00	0.94	4.00	1.10
(+)-Terpinen-4-ol	2.00	1.07	2.00	0.94	2.00	1.07	2.00	0.90	2.00	0.87	4.00	1.60
α-Terpineol	3.00	1.61	2.00	0.94	2.00	1.07	4.00	1.80	2.00	0.87	2.00	0.80
Thymol	1.00	2.29	0.50	1.11	0.50	1.14	0.50	1.10	0.75	1.64	0.75	1.61
Thymoquinone	2.00	1.24	2.00	1.11	2.00	1.24	2.00	1.07	2.00	1.04	2.00	0.97
Positive control	0.53		0.53		0.65		0.65		0.65		0.65	
Negative control	>8.00		3.33		>8.00		>8.00		>8.00		>8.00	
Culture control	>8.00		>8.00		>8.00		>8.00		>8.00		>8.00	

^1^ Noteworthy MIC values in bold; ^2^ antagonism in italics.

**Table 8 molecules-28-00030-t008:** Antimicrobial activity (MIC in mg/mL and ΣFIC) of the essential oil compound: carrier oil combinations against *Escherichia coli* ATCC 8739 (*n* = 3).

Compounds	Carrier Oils											
	*A. vera*		*C. officinalis*		*H. perforatum*		*P. americana*		*P. armeniaca*		*S. chinensis*	
	MIC	ΣFIC	MIC	ΣFIC	MIC	ΣFIC	MIC	ΣFIC	MIC	ΣFIC	MIC	ΣFIC
Carvacrol	1.33	1.29	1.80	1.80	1.33	1.33	1.33	1.33	1.33	1.26	1.33	1.26
β-Caryophyllene	4.00	1.27	4.00	1.40	4.00	1.40	2.00	0.70	4.00	1.17	16.00	*4.69*
Cinnamaldehyde	0.67	1.00	**0.67** ^1^	1.02	**0.67**	1.02	**0.83**	1.28	**0.67**	0.98	**0.67**	0.98
Citral	2.00	1.33	2.00	1.40	1.33	0.93	1.67	1.17	2.00	1.29	**1.00**	0.64
*p*-Cymene	4.00	1.33	4.00	1.47	4.00	1.47	2.67	0.98	4.00	1.24	16.00	*4.95*
Eugenol	2.00	1.33	2.00	1.40	2.67	1.87	2.00	1.40	2.00	1.29	2.00	1.29
Geraniol	2.00	1.17	1.67	1.03	1.67	1.03	2.00	1.23	2.00	1.12	1.67	0.93
Isoeugenol	1.33	0.89	1.33	0.93	1.33	0.93	1.33	0.93	1.67	1.07	**1.00**	0.64
R(+)-Limonene	4.00	1.04	4.00	1.18	4.00	1.18	4.00	1.18	4.00	0.95	4.00	0.95
Linalool	3.00	1.00	2.67	0.98	2.67	0.98	2.50	0.92	4.00	1.24	16.00	*4.95*
Linalyl acetate	4.00	1.33	3.33	1.22	3.33	1.22	16.00	5.87	4.00	1.24	4.00	1.24
Menthol	2.00	0.83	2.00	0.90	3.00	1.35	3.00	1.35	2.00	0.79	12.00	*4.71*
Nerol	2.00	0.93	3.33	1.67	3.33	1.67	2.67	1.33	2.00	0.89	2.00	0.89
(+)-α-Pinene	4.00	1.00	4.00	1.13	4.00	1.13	4.00	1.13	4.00	0.90	4.00	0.90
Santalol	3.00	3.50	4.00	*4.80* ^3^	4.00	*4.80*	2.00	2.40	2.00	2.29	2.00	2.29
α-Terpinene	6.00	1.90	3.33	1.17	4.00	1.40	2.00	0.70	16.00	*4.69*	4.00	1.17
γ-Terpinene	4.00	1.17	2.00	0.65	3.33	1.08	3.00	0.98	4.00	1.07	4.00	1.07
(+)-Terpinen-4-ol	4.00	1.47	2.00	0.80	2.67	1.07	4.00	1.60	4.00	1.37	4.00	1.37
α-Terpineol	3.00	1.25	2.00	0.90	3.00	1.35	2.00	0.90	3.00	1.18	3.00	1.18
Thymol	2.00	1.67	1.50	1.30	2.00	1.73	2.00	1.73	1.50	1.21	2.00	1.62
Thymoquinone	**0.02**	** *0.09* ** ^2^	**0.13**	0.51	**0.13**	0.51	**0.09**	** *0.38* **	**0.13**	** *0.50* **	**0.09**	** *0.37* **
Positive control	1.88		0.98		5.23		1.05		1.05		1.29	
Negative control	5.33		>8.00		5.33		5.33		5.33		>8.00	
Culture control	>8.00		>8.00		>8.00		>8.00		>8.00		>8.00	

^1^ Noteworthy MIC in bold; ^2^ synergy in bold and italics; ^3^ antagonism in italics.

**Table 9 molecules-28-00030-t009:** Antimicrobial activity (MIC in mg/mL and ΣFIC) of the essential oil compound: carrier oil combinations against *Candida albicans* ATCC 10231 (*n* = 3).

Compounds	Carrier Oils											
	*A. vera*		*C. officinalis*		*H. perforatum*		*P. americana*		*P. armeniaca*		*S. chinensis*	
	MIC	ΣFIC	MIC	ΣFIC	MIC	ΣFIC	MIC	ΣFIC	MIC	ΣFIC	MIC	ΣFIC
Carvacrol	**1.00** ^1^	2.50	**1.00**	2.50	**0.75**	1.75	**1.00**	2.17	**0.50**	1.13	**0.75**	1.88
β-Caryophyllene	1.50	1.13	2.00	1.50	**1.00**	0.58	2.00	0.83	**1.00**	*0.50*	1.50	1.13
Cinnamaldehyde	**0.13**	2.15	**0.13**	2.15	**0.13**	2.13	**0.13**	2.10	**0.13**	2.11	**0.13**	2.15
Citral	**0.50**	0.75	**0.38**	0.56	**0.50**	0.67	**0.50**	0.58	**0.50**	0.63	**0.50**	0.75
*p*-Cymene	**1.00**	0.83	**1.00**	0.83	**0.75**	** *0.50* ** ^2^	2.00	1.00	2.00	1.17	**1.00**	0.83
Eugenol	**1.00**	1.83	**0.75**	1.38	**0.50**	0.83	**0.50**	0.75	**0.50**	0.79	**0.50**	0.92
Geraniol	**0.75**	1.88	**1.00**	2.50	**0.50**	1.17	**0.50**	1.08	**0.50**	1.13	**0.50**	1.25
Isoeugenol	**1.00**	2.50	**0.50**	1.25	**0.50**	1.17	**0.50**	1.08	**0.50**	1.13	**0.50**	1.25
R(+)-Limonene	**1.00**	1.00	**1.00**	1.00	**1.00**	0.83	2.00	1.33	2.00	1.50	1.50	1.50
Linalool	**1.00**	1.00	**1.00**	1.00	3.00	2.50	**1.00**	0.67	1.50	1.13	**1.00**	1.00
Linalyl acetate	**1.00**	1.00	**1.00**	1.00	3.00	2.50	3.33	2.22	2.00	1.50	2.00	2.00
Menthol	**1.00**	1.50	**1.00**	1.50	2.00	2.67	**1.00**	1.17	**1.00**	1.25	**1.00**	1.50
Nerol	**1.00**	1.50	**1.00**	1.50	**1.00**	1.33	**1.00**	1.17	**1.00**	1.25	**1.00**	1.50
(+)-α-Pinene	**1.00**	1.00	**1.00**	1.00	2.00	1.67	1.50	1.00	1.50	1.13	**1.00**	1.00
Santalol	1.50	1.50	**1.00**	1.00	2.00	1.67	**1.00**	0.67	2.00	1.50	**1.00**	1.00
α-Terpinene	**1.00**	1.00	1.50	1.50	1.50	1.25	3.00	2.00	**1.00**	0.75	**1.00**	1.00
γ-Terpinene	3.00	2.25	**1.00**	0.75	**1.00**	0.58	3.00	1.25	3.00	1.50	**1.00**	0.75
(+)-Terpinen-4-ol	1.25	0.94	**1.00**	0.75	2.00	1.17	2.00	0.83	2.00	1.00	**1.00**	0.75
α-Terpineol	**1.00**	1.17	**1.00**	1.17	1.50	1.50	1.50	1.25	**1.00**	0.92	**1.00**	1.17
Thymol	**0.50**	0.75	**0.50**	0.75	**1.00**	1.33	**1.00**	1.17	**0.75**	0.94	**0.50**	0.75
Thymoquinone	**0.13**	0.86	**0.25**	1.73	**0.19**	1.26	**0.13**	0.82	**0.13**	0.83	**0.13**	0.90
Positive control	12.50		12.50		10.00		0.94		0.94		6.25	
Negative control	>8.00		>8.00		2.00		>8.00		>8.00		2.00	
Culture control	>8.00		>8.00		>8.00		>8.00		>8.00		>8.00	

^1^ Noteworthy MIC values in bold; ^2^ synergy in bold and italics.

**Table 10 molecules-28-00030-t010:** Toxicity (% mortality) and standard deviation (SD) at 24 and 48 h.

Sample	% Mortality (±SD)	
	24 h	48 h
*A. vera*	**1.13 (±1.60) ^1^**	**3.10 (±0.70)**
*C. officinalis*	**1.14 (±0.81)**	**1.14 (±0.81)**
*H. perforatum*	**0.69 (±0.98)**	**4.78 (±1.24)**
*P. americana*	**0.00 (±0.00)**	**0.95 (±1.35)**
*P. armeniaca*	**1.44 (±1.34)**	**3.38 (±1.67)**
*S. chinensis*	**2.27 (±3.21)**	**15.29 (±15.09)**
Carvacrol	100.00 (±0.00)	100.00 (±0.00)
β-Caryophyllene	**19.52 (±0.88)**	**41.65 (±1.66)**
Cinnamaldehyde	100.00 (±0.00)	100.00 (±0.00)
Citral	100.00 (±0.00)	100.00 (±0.00)
*p*-Cymene	**18.93 (±1.63)**	**38.24 (±7.42)**
Eugenol	100.00 (±0.00)	100.00 (±0.00)
Geraniol	100.00 (±0.00)	100.00 (±0.00)
Isoeugenol	100.00 (±0.00)	100.00 (±0.00)
R(+)-Limonene	**48.84 (±15.25)**	74.60 (±29.73)
Linalool	100.00 (±0.00)	100.00 (±0.00)
Linalyl acetate	**22.31 (±3.87)**	**27.73 (±3.08)**
Menthol	100.00 (±0.00)	100.00 (±0.00)
Nerol	100.00 (±0.00)	100.00 (±0.00)
(+)-α-Pinene	85.32 (±9.99)	94.70 (±3.53)
Santalol	100.00 (±0.00)	100.00 (±0.00)
α-Terpinene	57.19 (±7.30)	65.75 (±6.31)
γ-Terpinene	**9.86 (±11.23)**	**26.52 (±9.59)**
(+)-Terpinen-4-ol	100.00 (±0.00)	100.00 (±0.00)
α-Terpineol	100.00 (±0.00)	100.00 (±0.00)
Thymol	100.00 (±0.00)	100.00 (±0.00)
Thymoquinone	100.00 (±0.00)	100.00 (±0.00)
Potassium dichromate (positive control)	100.00 (±0.00)	100.00 (±0.00)
2% DMSO (negative control)	**0.41 (±0.57)**	**1.69 (±1.60)**
20% Diluted acetone (negative control) ^2^	**1.85 (±0.41)**	**7.53 (±1.60)**
50% Diluted acetone (negative control)	**1.90(±0.19)**	**15.04(±3.97)**
Salt water (negative control)	**3.45 (±2.53)**	**6.51 (±1.22)**

^1^ Bold values represent biological non-toxicity; shaded area shows carrier oils and non-shaded area shows compounds. ^2^ Although acetone is known as a toxic agent to brine shrimp, it was the only solvent that allowed dilution of several insoluble compounds; thus, diluted acetone was used and included as a negative control.

**Table 11 molecules-28-00030-t011:** Mean toxicity (% mortality), standard deviation (SD), ƩFPM (fractional percentage mortality index), and interpretation of essential oil compound: *Aloe vera* combinations (*n* = 3).

Essential Oil Compound	*Aloe vera*							
	24 h				48 h			
	% Mortality (±SD)	Incr/Decr Toxicity ^1^	ƩFIC	Int ^2^	% Mortality (±SD)	Incr/Decr Toxicity	ƩFIC	Int
Carvacrol	100.00 (±0.00)	Equal	*44.75*	Ant	100.00 (±0.00)	Equal	*16.63*	Ant
β-Caryophyllene	**3.85 (±3.23)** ^3^	5-fold decr	1.80	Ind	**38.17 (±9.09)**	1-fold decr	*6.61*	Ant
Cinnamaldehyde	100.00 (±0.00)	Equal	*44.75*	Ant	100.00 (±0.00)	Equal	*16.63*	Ant
Citral	100.00 (±0.00)	Equal	*44.75*	Ant	100.00 (±0.00)	Equal	*16.63*	Ant
*p*-Cymene	**0.61 (±0.86)**	31-fold decr	** *0.28* **	**Syn**	**6.90 (±6.73)**	6-fold decr	1.20	Ind
Eugenol	100.00 (±0.00)	Equal	*44.75*	Ant	100.00 (±0.00)	Equal	*16.63*	Ant
Geraniol	100.00(±0.00)	Equal	*44.75*	Ant	100.00 (±0.00)	Equal	*16.63*	Ant
Isoeugenol	**38.96 (±10.70)**	3-fold decr	*17.44*	Ant	75.88 (±13.85)	1-fold decr	*12.62*	Ant
R(+)-Limonene	81.25 (±9.56)	2-fold incr	*36.78*	Ant	92.25 (±1.51)	1-fold incr	*15.5*	Ant
Linalool	63.17 (±5.09)	2-fold decr	*28.27*	Ant	90.32 (±1.78)	1-fold decr	*15.02*	Ant
Linalyl acetate	**1.67 (±2.36)**	13-fold decr	0.77	Add	**4.01 (±1.81)**	7-fold decr	0.72	Add
Menthol	100.00 (±0.00)	Equal	*44.75*	Ant	100.00 (±0.00)	Equal	*16.63*	Ant
Nerol	94.62 (±2.57)	1-fold decr	*42.34*	Ant	99.42 (±0.83)	1-fold decr	*16.53*	Ant
(+)-α-Pinene	95.88 (±0.37)	1-fold incr	*42.99*	Ant	97.85 (±1.70)	1-fold incr	*16.30*	Ant
Santalol	**18.74 (±10.75)**	5-fold decr	*8.39*	Ant	69.29 (±11.64)	1-fold decr	*11.52*	Ant
α-Terpinene	**0.81 (±1.15)**	71-fold decr	** *0.37* **	**Syn**	**2.88 (±2.09)**	23-fold decr	** *0.49* **	**Syn**
γ-Terpinene	100.00 (±0.00)	10-fold incr	*49.32*	Ant	100.00 (±0.00)	4-fold incr	*18.01*	Ant
(+)-Terpinen-4-ol	90.38 (±6.46)	1-fold decr	*40.44*	Ant	96.33 (±3.81)	1-fold decr	*16.02*	Ant
α-Terpineol	100.00 (±0.00)	Equal	*44.75*	Ant	100.00 (±0.00)	Equal	*16.63*	Ant
Thymol	100.00 (±0.00)	Equal	*44.75*	Ant	100.00 (±0.00)	Equal	*16.63*	Ant
Thymoquinone	100.00 (±0.00)	Equal	*44.75*	Ant	100.00 (±0.00)	Equal	*16.63*	Ant

^1^ Incr/decr toxicity (increase/decrease toxicity)—the increase or decrease in toxicity of the compound from when tested alone to when combined with the carrier oil. ^2^ Int (interpretation)—the interpretation of ƩFIC values, whether antagonistic (Ant) (italics), synergistic (Syn) (bold and italics), additive (Add), or indifferent (Ind). ^3^ Bold values represent biological non-toxicity.

**Table 12 molecules-28-00030-t012:** Mean toxicity (% mortality), standard deviation (SD), ƩFPM, and interpretation of essential oil compound: *Calendula officinalis* combinations (*n* = 3).

Essential Oil Compound	*Calendula officinalis*							
	24 h				48 h			
	% Mortality (±SD)	Incr/Decr Toxicity ^1^	ƩFIC	Int ^2^	% Mortality (±SD)	Incr/Decr Toxicity	ƩFIC	Int
Carvacrol	100.00 (±0.00)	Equal	*44.36*	Ant	100.00 (±0.00)	Equal	*44.36*	Ant
β-Caryophyllene	**5.70 (±5.22)** ^3^	3-fold decr	2.65	Ind	**40.49 (±22.73)**	1-fold decr	*18.25*	Ant
Cinnamaldehyde	100.00 (±0.00)	Equal	*44.36*	Ant	100.00 (±0.00)	Equal	*44.36*	Ant
Citral	100.00 (±0.00)	Equal	*44.36*	Ant	100.00 (±0.00)	Equal	*44.36*	Ant
*p*-Cymene	**37.12 (±2.59)**	2-fold incr	*17.26*	Ant	83.69 (±14.40)	2-fold incr	*37.80*	Ant
Eugenol	100.00 (±0.00)	Equal	*44.36*	Ant	100.00 (±0.00)	Equal	*44.36*	Ant
Geraniol	100.00(±0.00)	Equal	*44.36*	Ant	100.00 (±0.00)	Equal	*44.36*	Ant
Isoeugenol	**21.40 (±6.96)**	5-fold decr	*9.49*	Ant	89.86 (±13.05)	1-fold decr	*39.86*	Ant
R(+)-Limonene	**0.00 (±0.00)**	Complete decr	**NV ^4^**	**Syn**	**8.15 (±3.98)**	9-fold decr	3.63	Ind
Linalool	**6.66 (±1.39)**	15-fold decr	2.95	Ind	70.46 (±17.83)	1-fold decr	*31.26*	Ant
Linalyl acetate	**0.69 (±0.50)**	32-fold decr	** *0.32* **	**Syn**	**8.34 (±1.76)**	3-fold decr	3.81	Ind
Menthol	100.00 (±0.00)	Equal	*44.36*	Ant	100.00 (±0.00)	Equal	*44.36*	Ant
Nerol	95.25 (±0.94)	1-fold decr	*42.25*	Ant	100.00 (±0.00)	Equal	*44.36*	Ant
(+)-α-Pinene	**4.36 (±3.11)**	20-fold decr	1.94	Ind	79.14 (±9.01)	1-fold decr	*35.13*	Ant
Santalol	78.13 (±0.50)	1-fold decr	*34.66*	Ant	99.61 (±0.55)	1-fold decr	*44.19*	Ant
α-Terpinene	**24.58 (±5.37)**	2-fold decr	*11.00*	Ant	65.62 (±8.55)	1-fold decr	*29.28*	Ant
γ-Terpinene	**0.00 (±0.00)**	Complete decr	**NV**	**Syn**	**0.56 (±0.79)**	47-fold decr	** *0.25* **	**Syn**
(+)-Terpinen-4-ol	**0.48 (±0.67)**	208-fold decr	** *0.21* **	**Syn**	**15.26 (±3.89)**	7-fold decr	*6.77*	Ant
α-Terpineol	100.00 (±0.00)	Equal	*44.36*	Ant	100.00 (±0.00)	Equal	*44.36*	Ant
Thymol	100.00 (±0.00)	Equal	*44.36*	Ant	100.00 (±0.00)	Equal	*44.36*	Ant
Thymoquinone	100.00 (±0.00)	Equal	*44.36*	Ant	100.00 (±0.00)	Equal	*44.36*	Ant

^1^ Incr/decr toxicity (increase/decrease toxicity)—the increase or decrease in toxicity of the compound from when tested alone to when combined with the carrier oil. ^2^ Int (interpretation)—the interpretation of ƩFIC values, whether antagonistic (Ant) (italics), synergistic (Syn) (bold and italics), additive (Add), or indifferent (Ind). ^3^ Bold values represent biological non-toxicity. ^4^ NV—no value could be calculated due to the carrier oil’s toxicity being 0.00%.

**Table 13 molecules-28-00030-t013:** Mean toxicity (% mortality), standard deviation (SD), ƩFPM, and interpretation of essential oil compound: *Hypericum perforatum* combinations (*n* = 3).

Essential Oil Compound	*Hypericum perforatum*							
	24 h				48 h			
	% Mortality (±SD)	Incr/Decr Toxicity ^1^	ƩFIC	Int ^2^	% Mortality (±SD)	Incr/Decr Toxicity	ƩFIC	Int
Carvacrol	100.00 (±0.00)	Equal	*72.96*	Ant	100.00 (±0.00)	Equal	*10.96*	Ant
β-Caryophyllene	**4.57 (±1.55)** ^3^	4-fold decr	3.43	Ind	**25.18 (±3.52)**	2-fold decr	2.94	Ind
Cinnamaldehyde	100.00 (±0.00)	Equal	*72.96*	Ant	100.00 (±0.00)	Equal	*10.96*	Ant
Citral	100.00 (±0.00	Equal	*72.96*	Ant	100.00 (±0.00)	Equal	*10.96*	Ant
*p*-Cymene	**22.42 (±9.18)**	1-fold incr	*16.84*	Ant	61.16 (±18.45)	2-fold incr	*7.20*	Ant
Eugenol	100.00 (±0.00)	Equal	*72.96*	Ant	100.00 (±0.00)	Equal	*10.96*	Ant
Geraniol	100.00 (±0.00)	Equal	*72.96*	Ant	100.00 (±0.00)	Equal	*10.96*	Ant
Isoeugenol	**46.84 (±20.84)**	2-fold decr	*34.17*	Ant	91.62 (±4.65)	1-fold decr	*10.04*	Ant
R (+)-Limonene	**3.09 (±3.73)**	16-fold decr	2.27	Ind	**4.04 (±3.41)**	18-fold decr	** *0.45* **	**Syn**
Linalool	**18.59 (±6.49)**	5-fold decr	*13.57*	Ant	86.17 (±3.85)	1-fold decr	*9.44*	Ant
Linalyl acetate	**1.79 (±1.75)**	12-fold decr	1.34	Ind	**7.12 (±3.64)**	4-fold decr	0.87	Add
Menthol	100.00 (±0.00)	Equal	*72.96*	Ant	100.00 (±0.00)	Equal	*10.96*	Ant
Nerol	96.48 (±1.63)	1-fold decr	*70.4*	Ant	100.00 (±0.00)	Equal	*10.96*	Ant
(+)-α-Pinene	71.13 (±1.77)	1-fold decr	*51.96*	Ant	98.66 (±0.14)	1-fold incr	*10.84*	Ant
Santalol	**0.63 (±0.89)**	159-fold decr	** *0.46* **	**Syn**	**27.44 (±5.18)**	4-fold decr	3.01	Ind
α-Terpinene	**11.15 (±4.76)**	5-fold decr	*8.18*	Ant	**33.33 (±8.26)**	2-fold decr	3.74	Ind
γ-Terpinene	**0.68 (±0.96)**	15-fold decr	0.53	Add	**3.42 (±1.07)**	8-fold decr	** *0.42* **	**Syn**
(+)-Terpinen-4-ol	**1.98 (±1.67)**	51-fold decr	1.44	Ind	**13.49 (±2.97)**	7-fold decr	1.48	Ind
α-Terpineol	100.00 (±0.00)	Equal	*72.96*	Ant	100.00 (±0.00)	Equal	*10.96*	Ant
Thymol	100.00 (±0.00)	Equal	*72.96*	Ant	100.00 (±0.00)	Equal	*10.96*	Ant
Thymoquinone	100.00 (±0.00)	Equal	*72.96*	Ant	100.00 (±0.00)	Equal	*10.96*	Ant

^1^ Incr/decr toxicity (increase/decrease toxicity)—the increase or decrease in toxicity of the compound from when tested alone to when combined with the carrier oil. ^2^ Int (interpretation)—the interpretation of ƩFIC values, whether antagonistic (Ant) (italics), synergistic (Syn) (bold and italics), additive (Add), or indifferent (Ind). ^3^ Bold values represent biological non-toxicity.

**Table 14 molecules-28-00030-t014:** Mean toxicity (% mortality), standard deviation (SD), ƩFPM, and interpretation of essential oil compound: *Persea americana* combinations (*n* = 3).

Essential Oil Compound	*Persea americana*							
	24 h				48 h			
	% Mortality (±SD)	Incr/Decr Toxicity ^1^	ƩFIC	Int ^2^	% Mortality (±SD)	Incr/Decr Toxicity	ƩFIC	Int
Carvacrol	100.00 (±0.00)	Equal	- ^4^	-	100.00 (±0.00)	Equal	*53.13*	Ant
β-Caryophyllene	**10.96 (±9.23)** ^3^	2-fold decr	-	-	50.12 (±14.10)	1-fold incr	*26.98*	Ant
Cinnamaldehyde	100.00 (±0.00)	Equal	-	-	100.00 (±0.00)	Equal	*53.13*	Ant
Citral	100.00 (±0.00)	Equal	-	-	100.00 (±0.00)	Equal	*53.13*	Ant
*p*-Cymene	**37.79 (±5.23)**	2-fold incr	-	-	**48.19 (±1.08)**	1-fold incr	*25.99*	Ant
Eugenol	100.00 (±0.00)	Equal	-	-	100.00 (±0.00)	Equal	*53.13*	Ant
Geraniol	100.00 (±0.00)	Equal	-	-	100.00 (±0.00)	Equal	*53.13*	Ant
Isoeugenol	**47.69 (±21.04)**	2-fold decr	-	-	82.80 (±10.15)	1-fold decr	*44.00*	Ant
R(+)-Limonene	**48.76 (±4.70)**	1-fold decr	-	-	66.84 (±7.72)	1-fold decr	*35.63*	Ant
Linalool	**17.20 (±10.04)**	6-fold decr	-	-	89.16 (±4.60)	1-fold decr	*47.37*	Ant
Linalyl acetate	**6.40 (±2.73)**	3-fold decr	-	-	**22.59 (±7.30)**	1-fold decr	*12.30*	Ant
Menthol	100.00 (±0.00)	Equal	-	-	100.00 (±0.00)	Equal	*53.13*	Ant
Nerol	100.00 (±0.00)	Equal	-	-	100.00 (±0.00)	Equal	*53.13*	Ant
(+)-α-Pinene	79.20 (±5.30)	1-fold decr	-	-	98.37 (±1.36)	1-fold incr	*52.29*	Ant
Santalol	**1.38 (±1.12)**	72-fold decr	-	-	**2.78 (±1.97)**	36-fold decr	1.48	Ind
α-Terpinene	**3.12 (±3.46)**	18-fold decr	-	-	62.95 (±18.31)	1-fold decr	*33.61*	Ant
γ-Terpinene	100.00 (±0.00)	10-fold incr	-	-	100.00 (±0.00)	4-fold incr	*54.52*	Ant
(+)-Terpinen-4-ol	**3.69 (±2.50)**	27-fold decr	-	-	**48.41 (±7.13)**	2-fold decr	*25.72*	Ant
α-Terpineol	100.00 (±0.00)	Equal	-	-	100.00 (±0.00)	Equal	*53.13*	Ant
Thymol	100.00 (±0.00)	Equal	-	-	100.00 (±0.00)	Equal	*53.13*	Ant
Thymoquinone	100.00 (±0.00)	Equal	-	-	100.00 (±0.00)	Equal	*53.13*	Ant

^1^ Incr/decr toxicity (increase/decrease toxicity)—the increase or decrease in toxicity of the compound from when tested alone to when combined with the carrier oil. ^2^ Int (interpretation)—the interpretation of ƩFIC values, whether antagonistic (Ant) (italics), synergistic (Syn) (bold and italics), additive (Add), or indifferent (Ind). ^3^ Bold values represent biological non-toxicity. ^4^ Value could not be calculated due to the carrier oil’s toxicity being 0.00%.

**Table 15 molecules-28-00030-t015:** Mean toxicity (% mortality), standard deviation (SD), ƩFPM, and interpretation of essential oil compound: *Prunus armeniaca* combinations (n = 3).

Essential Oil Compound	*Prunus armeniaca*							
	24 h				48 h			
	% Mortality (±SD)	Incr/Decr Toxicity ^1^	ƩFIC	Int ^2^	% Mortality (± SD)	Incr/Decr Toxicity	ƩFIC	Int
Carvacrol	100.00 (±0.00)	Equal	*35.22*	Ant	100.00 (±0.00)	Equal	*15.29*	Ant
β-Caryophyllene	**13.56 (±11.70)** ^3^	1-fold decr	*5.06*	Ant	**42.37 (±20.55)**	1-fold incr	*6.78*	Ant
Cinnamaldehyde	100.00 (±0.00)	Equal	*35.22*	Ant	100.00 (±0.00)	Equal	*15.29*	Ant
Citral	100.00 (±0.00)	Equal	*35.22*	Ant	100.00 (±0.00)	Equal	*15.29*	Ant
*p*-Cymene	52.07 (±11.54)	3-fold incr	*19.46*	Ant	65.88 (±8.12)	2-fold incr	*10.61*	Ant
Eugenol	100.00 (±0.00)	Equal	*35.22*	Ant	100.00 (±0.00)	Equal	*15.29*	Ant
Geraniol	100.00 (±0.00)	Equal	*35.22*	Ant	100.00 (±0.00)	Equal	*15.29*	Ant
Isoeugenol	**9.31 (±8.72)**	11-fold decr	3.28	Ind	58.40 (±5.45)	2-fold decr	*8.93*	Ant
R(+)-Limonene	**0.55 (±0.77)**	89-fold decr	** *0.20* **	**Syn**	**1.26 (±0.91)**	59-fold decr	** *0.19* **	**Syn**
Linalool	**26.64 (±3.99)**	4-fold decr	*9.38*	Ant	90.97 (±5.88)	1-fold decr	*13.91*	Ant
Linalyl acetate	**0.78 (±1.11)**	29-fold decr	** *0.29* **	**Syn**	**1.99 (±1.99)**	14-fold decr	** *0.33* **	**Syn**
Menthol	100.00 (±0.00)	Equal	*35.22*	Ant	100.00 (±0.00)	Equal	*15.29*	Ant
Nerol	100.00 (±0.00)	Equal	*35.22*	Ant	100.00 (±0.00)	Equal	*15.29*	Ant
(+)-α-Pinene	**32.97 (±17.92)**	3-fold decr	*11.64*	Ant	68.70 (±20.45)	1-fold decr	*10.53*	Ant
Santalol	**0.00 (±0.00)**	Complete decr	**NV** ^4^	**Syn**	**1.26 (±0.89)**	79-fold decr	** *0.19* **	**Syn**
α-Terpinene	**5.98 (±3.09)**	10-fold decr	2.13	Ind	71.18 (±18.94)	1-fold incr	*11.07*	Ant
γ-Terpinene	**5.64 (±2.83)**	2-fold decr	2.25	Ind	**20.61 (±1.75)**	1-fold decr	3.44	Ind
(+)-Terpinen-4-ol	**7.51 (±3.83)**	13-fold decr	2.64	Ind	65.88 (±4.68)	2-fold decr	*10.08*	Ant
α-Terpineol	100.00 (±0.00)	Equal	*35.22*	Ant	100.00 (±0.00)	Equal	*15.29*	Ant
Thymol	100.00 (±0.00)	Equal	*35.22*	Ant	100.00 (±0.00)	Equal	*15.29*	Ant
Thymoquinone	100.00 (±0.00)	Equal	*35.22*	Ant	100.00 (±0.00)	Equal	*15.29*	Ant

^1^ Incr/decr toxicity (increase/decrease toxicity)—the increase or decrease in toxicity of the compound from when tested alone to when combined with the carrier oil. ^2^ Int (interpretation)—the interpretation of ƩFIC values, whether antagonistic (Ant) (italics), synergistic (Syn) (bold and italics), additive (Add), or indifferent (Ind). ^3^ Bold values represent biological non-toxicity. ^4^ NV—no value could be calculated as the % mortality of the combination was 0.00.

**Table 16 molecules-28-00030-t016:** Mean toxicity (% mortality), standard deviation (SD), ƩFPM, and interpretation of essential oil compound: *Simmondsia chinensis* combinations (*n* = 3).

Essential Oil Compound	*Simmondsia chinensis*							
	24 h				48 h			
	% Mortality (±SD)	Incr/Decr Toxicity ^1^	ƩFIC	Int ^2^	% Mortality (±SD)	Incr/Decr Toxicity	ƩFIC	Int
Carvacrol	100.00 (±0.00)	Equal	*22.53*	Ant	100.00 (±0.00)	Equal	3.77	Ind
β-Caryophyllene	**16.26 (±4.95)** ^3^	1-fold decr	*4.00*	Ant	68.47 (±5.74)	2-fold incr	3.06	Ind
Cinnamaldehyde	100.00 (±0.00)	Equal	*22.53*	Ant	100.00 (±0.00)	Equal	3.77	Ind
Citral	100.00 (±0.00)	Equal	*22.53*	Ant	100.00 (±0.00)	Equal	3.77	Ind
*p*-Cymene	86.56 (±2.28)	5-fold incr	*21.35*	Ant	92.48 (±3.14)	2-fold incr	*4.23*	Ant
Eugenol	100.00 (±0.00)	Equal	*22.53*	Ant	100.00 (±0.00)	Equal	3.77	Ind
Geraniol	100.00 (±0.00)	Equal	*22.53*	Ant	100.00 (±0.00)	Equal	3.77	Ind
Isoeugenol	70.84 (±17.88)	1-fold decr	*15.96*	Ant	96.16 (±2.83)	1-fold decr	3.63	Ind
R(+)-Limonene	67.02 (±24.02)	1-fold incr	*15.45*	Ant	85.67 (±13.21)	1-fold incr	3.38	Ind
Linalool	100.00 (±0.00)	Equal	*22.53*	Ant	100.00 (±0.00)	Equal	3.77	Ind
Linalyl acetate	88.41 (±16.40)	4-fold incr	*21.45*	Ant	96.38 (±5.12)	3-fold incr	*4.89*	Ant
Menthol	85.71 (±5.73)	1-fold decr	*19.31*	Ant	100.00 (±0.00)	Equal	3.77	Ind
Nerol	100.00 (±0.00)	Equal	*22.53*	Ant	100.00 (±0.00)	Equal	3.77	Ind
(+)-α-Pinene	100.00 (±0.00)	1-fold incr	*22.61*	Ant	100.00 (±0.00)	1-fold incr	3.80	Ind
Santalol	100.00 (±0.00)	Equal	*22.53*	Ant	100.00 (±0.00)	Equal	3.77	Ind
α-Terpinene	72.51 (±14.26)	1-fold incr	*16.61*	Ant	87.89 (±10.59)	1-fold incr	3.54	Ind
γ-Terpinene	71.27 (±9.04)	7-fold incr	*19.31*	Ant	79.51 (±5.71)	3-fold incr	*4.10*	Ant
(+)-Terpinen-4-ol	98.92 (±1.52)	1-fold decr	*22.28*	Ant	100.00 (±0.00)	Equal	3.77	Ind
α-Terpineol	100.00 (±0.00)	Equal	*22.53*	Ant	100.00 (±0.00)	Equal	3.77	Ind
Thymol	100.00 (±0.00)	Equal	*22.53*	Ant	100.00 (±0.00)	Equal	3.77	Ind
Thymoquinone	100.00 (±0.00)	Equal	*22.53*	Ant	100.00 (±0.00)	Equal	3.77	Ind

^1^ Incr/decr toxicity (increase/decrease toxicity)—the increase or decrease in toxicity of the compound from when tested alone to when combined with the carrier oil. ^2^ Int (interpretation)—the interpretation of ƩFIC values, whether antagonistic (Ant) (italics), synergistic (Syn) (bold and italics), additive (Add), or indifferent (Ind). ^3^ Bold values represent biological non-toxicity.

**Table 17 molecules-28-00030-t017:** Interactions (%) of each carrier oil within its respective compound–carrier oil combinations.

Carrier Oil	24 h				48 h			
	% Syn ^1^	% Ant ^2^	% Ind ^3^	% Add ^4^	% Syn	% Ant	% Ind	% Add
*A. vera*	10	80	5	5	5	85	5	5
*C. officinalis*	19	67	14	0	5	85	10	0
*H. perforatum*	5	71	19	5	10	66	19	5
*P. americana*	-	-	-	-	0	95	5	0
*P. armeniaca*	14	72	19	0	14	81	5	0
*S. chinensis*	0	100	0	0	0	14	86	0

^1^ % synergy; ^2^ % antagonism; ^3^ % indifference; ^4^ % additive.

**Table 18 molecules-28-00030-t018:** Selectivity index (SI) of synergistic combinations found in the antimicrobial studies.

Carrier Oil	Compound	SI at 24 h	SI at 48 h
** *E. faecium* **			
*A. vera*	Menthol	0	0
*A. vera*	α-Terpinene	**15** ^1^	4
*C. officinalis*	α-Terpinene	1	0
*H. perforatum*	α-Terpinene	1	0
** *S. aureus* **			
*A. vera*	Thymoquinone	**500**	**500**
*C. officinalis*	Thymoquinone	**167**	**167**
*H. perforatum*	Thymoquinone	**500**	**500**
*P. americana*	Geraniol	1	1
*P. americana*	Thymol	1	1
*P. americana*	Thymoquinone	**167**	**167**
*P. armeniaca*	Eugenol	1	1
*P. armeniaca*	Geraniol	1	1
*P. armeniaca*	Thymol	1	1
*P. armeniaca*	Thymoquinone	**500**	**500**
*S. chinensis*	Eugenol	1	1
*S. chinensis*	Geraniol	1	1
*S. chinensis*	Thymoquinone	**250**	**250**
** *E. coli* **			
*A. vera*	Thymoquinone	**25**	**25**
*P. americana*	Thymoquinone	**6**	**6**
*P. armeniaca*	Thymoquinone	4	4
*S. chinensis*	Thymoquinone	**6**	**6**
** *C. albicans* **			
*H. perforatum*	p-Cymene	3	1
*P. armeniaca*	β-Caryophyllene	4	1

^1^ Bold values represent acceptable SI values.

## Data Availability

Not applicable.

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
