# Peer review of "The Antimicrobial and Toxicity Influence of Six Carrier Oils on Essential Oil Compounds"

_molecules, 2022, doi:10.3390/molecules28010030_

Round 1
Reviewer 1 Report
This manuscript represents a valuable contribution to the field, describing a complex research that brings new data on the antimicrobial and toxicity effects of carrier oils in combination with essential oil compounds to determine which combinations would provide the optimum antimicrobial combination with the least toxicity.
The issues addressed are timely and aim to fill some of the existing gaps regarding both the reduction of essential oils toxicity and to provide information on interactions that may occur when adding a carrier oil to essential oil compounds.
The paper is well introduced, the materials and methods are well detailed, and the results and discussion are well presented and explained. The conclusions are clearly stated and supported by the results. The references are written in agreement with the journal requirements, being relevant for the research topic.
Although the work has no has no conceptual and scientific weaknesses, an editing review is needed. The content of this manuscript fits well with the journal topic and therefore, I recommend its publication after a minor revision.

Author Response
Thank you very much for the positive response to the manuscript. As requested, some editing requirements as stipulated in the PDF are required. These have been attended to as detailed in the attached list of corrections

Reviewer 2 Report
The paper is well written and provides valuable information. The paper contains huge data which provides outstanding results. Some comments are raised please see the attached PDF

Author Response
Thank you for the feedback and the reviewer's valuable time to evaluate the manuscript. Comments made by the reviewer's in the provided PDF have been addressed and can be found in the attached;

Reviewer 3 Report
The research paper “The antimicrobial and toxicity influence of carrier oils on essential oil compounds” by Salehah Moola, Ané Orchard, and Sandy van Vuuren discusses that Essential oils are potent antimicrobials and carrier oils in combination with essential oil reduce its toxicity.
The study defines interesting research on tackling the challenges associated with Essential oils and reduce its toxicity, when used in combination.
The antimicrobial properties were evaluated against multiple pathogens namely ESKAPE pathogens, presently emerging drug-resistant strains against antimicrobials.
While, a previous study in similar area (Orchard et al, 2019) studied the combination of essential oils with carrier oil and its impact on Antimicrobial Activity and Cytotoxicity of Essential Oil, the present study further discussed the presence of single bioactive components in essential oil.
The paper discuss interesting findings and potential of carrier oils in reducing toxicity of essential oils as antimicorbials, highlighting that this combination can be further explored in countering drug-resistant pathogens.
Author Response
Thank you to the reviewer who took the time to evaluate our manuscript. No further amendments were requested.
Reviewer 4 Report
Despite the huge amount of data of the manuscript, it is still difficult to identify the innovation. The huge data made the article appears to have a lot of redundant information. The article seems more like a phenomenon description without in-depth research. In my view, the current formulation of the manuscript is not very suitable for publishing in molecules.
Author Response
Thank you for the time taken to evaluate our manuscript. I am sorry that the innovation did not come through as much as it possibly should have. The novelty lies in the following;
- Over 880 combinations (essential oil compounds with carrier oils) were undertaken in the antimicrobial studies. To the best of our knowledge this has been studied previously.
- The compound thymoquinone and the carrier oil P. armeniaca were present in the majority of the synergistic combinations-Never been demonstrated previously.
- The carrier oil H. perforatum and the compound santalol were present most often in antagonistic combinations-Never been demonstrated previously.
- The investigation of 126 combinations within the toxicity studies have not been previously studied.
- The toxicity of several compounds such as R (+)-limonene, santalol and γ- terpinene, was reduced to non-toxic levels when combined with carrier oils-Never been demonstrated previously.
- The compound thymoquinone was present in majority (> 50%) of the combinations which had an SI value > 4, showing that the antimicrobial activity of the compound: carrier oil combination was not due to toxicity--Never been demonstrated previously.
Round 2
Reviewer 4 Report
-
The work was complex and specific, and the experiment was well designed. I recommend to accept.